# Nano-Spray-Dried Levocetirizine Dihydrochloride with Mucoadhesive Carriers and Cyclodextrins for Nasal Administration

**DOI:** 10.3390/pharmaceutics15020317

**Published:** 2023-01-18

**Authors:** Mirella Mirankó, Judit Tóth, Csilla Bartos, Rita Ambrus, Tivadar Feczkó

**Affiliations:** 1Institute of Materials and Environmental Chemistry, Research Centre for Natural Sciences, Magyar Tudósok Körútja 2, 1117 Budapest, Hungary; 2Faculty of Engineering, Research Institute of Biomolecular and Chemical Engineering, University of Pannonia, Egyetem u. 10, 8200 Veszprém, Hungary; 3Faculty of Pharmacy, Institute of Pharmaceutical Technology and Regulatory Affairs, University of Szeged, Eötvös Street 6, 6720 Szeged, Hungary

**Keywords:** levocetirizine, nano spray drying, nasal administration, cyclodextrin, polymeric excipient

## Abstract

Antihistamines such as levocetirizine dihydrochloride (LC) are commercially used in oral tablets and oral drops to reduce allergic symptoms. In this study, LC was nano-spray-dried using three mucoadhesive polymers and four cyclodextrin species to form composite powders for nasal administration. The product was composed of hydroxypropyl methylcellulose polymer, including LC as a zwitterion, after neutralization by NaOH, and XRD investigations verified its amorphous state. This and a sulfobutylated-beta-cyclodextrin sodium salt-containing sample showed crystal peaks due to NaCl content as products of the neutralization reaction in the solutions before drying. The average particle size of the spherical microparticles was between 2.42 and 3.44 µm, except for those containing a polyvinyl alcohol excipient, which were characterized by a medium diameter of 29.80 µm. The drug was completely and immediately liberated from all the samples at pH 5.6 and 32 °C; i.e., the carriers did not change the good dissolution behavior of LC. A permeability test was carried out by dipping the synthetic cellulose ester membrane in isopropyl myristate using modified horizontal diffusion cells. The spray-dried powder with β-cyclodextrin showed the highest permeability (188.37 µg/cm^2^/h), as this additive was the least hydrophilic. Products prepared with other cyclodextrins (randomly methylated-beta-cyclodextrin, sulfobutylated-beta-cyclodextrin sodium salt and (hydroxypropyl)-beta-cyclodextrin) showed similar or slightly higher penetration abilities than LC. Other polymer excipients resulted in lower penetration of the active agent than the pure LC.

## 1. Introduction

Many people suffer from pollinosis or allergic rhinitis. Intranasal corticosteroids and also antihistamines can be used to reduce allergic symptoms; nevertheless, physicians prefer antihistamines because of the adverse effects of steroids [1].

Histamine is a major chemical mediator that causes nasal allergy symptoms through its action on the histamine H1 receptor [2]. Antihistamines decrease nasal symptoms by blocking the effect of histamine on the H1 receptor [3]. Two types of antihistamines can be distinguished: neutral antagonists and inverse agonists of H1R. Neutral antagonists obstruct histamine by blocking its binding on H1R. The inverse agonists obstruct histamine on H1R and suppress constitutive H1R activity in the absence of histamine [4,5].

Oral antihistamines are commonly used to decrease allergic symptoms; however, nasal preparations are also available. There are many advantages to using the nasal route, i.e., avoidance of liver first-pass metabolism, metabolism of the gastrointestinal tract and decreased chance of overdose [6]; thus, intranasal administration shows better systemic tolerability [7]. In the case of seasonal allergic rhinitis, oral cetirizine tablets and an azelastine nasal spray were investigated by Berger et al. [8]. Both antihistamines were effective at treating nasal symptoms, but the azelastine nasal spray performed better in the case of nasal congestion and sneezing due to the greater local concentration of the drug in the nasal mucosa. The disadvantage of the spray is that some of the people perceived a bitter taste. The systemic adverse effects may be decreased via intranasal administration, which are more frequent in case of oral administration [9]. Other studies investigated the effectiveness of oral antihistamines, such as loratadine and fexofenadine, against azelastine nasal spray. Even in these cases, the nasal antihistamine performed better in treating seasonal allergic rhinitis [10,11]. Olopatadine histamine was studied as a tablet and as a nasal agent. Nasal agents differ from their oral counterparts as they have a significant effect on nasal congestion, which is an annoying symptom [12].

Powder-based nasal formulations are not yet widespread; nevertheless, some studies have shown that they provide better results than liquid-based nasal sprays [13]. Rapid clearance from the nasal cavity is a significant disadvantage of nasal fluids, but excipients may increase the stability of the composition in the case nasal powders are used [14,15]. Additives of nasal powders allow higher doses of the drug to be administered and enhance the diffusion and absorption of the drug across the mucosa, improving its bioavailability relative to nasal fluids [16,17]. Additives may have different roles in nasal powders, such as fillers, absorption enhancers and mucoadhesive materials [16]. Some polymers can behave as fillers and mucoadhesive agents at the same time. When the polymer-containing particles enter the nose, the polymer chains are hydrated, while nasal secretion is dehydrated; thus, a viscous substance is created in the nose, prolonging the residence time of the drug [18]. In the case of cellulose derivatives such as mucoadhesive hydroxypropyl cellulose [19], it was proved that there is a connection between nasal absorption and the solubility/permeability of the drug [20]. When mucoadhesive polyvinylpyrrolidone or polyvinyl alcohol [21] was used, the absorption to the mucosa was enhanced. Moreover, the polymer matrix did not delay drug release; thus, the effect was rapid [22]. Absorption enhancers such as cyclodextrins and their derivatives can alter permeability, thereby improving the absorption [23,24]. Cyclodextrin has significant potential to mask the bitter taste of cetirizine and to hinder the instability induced by oxidizing factors [25]. The additives in powder-based nasal products can be physically mixed with the drug [26], co-lyophilized or co-spray-dried with the active agent [27,28,29,30]; furthermore, the drug can be dispersed, forming a matrix, [31] or a core–shell structure may be produced in the microcapsules or liposomes [32,33]. There are several methods used to manufacture nasal powders, such as precipitation [34], solvent evaporation processes [35,36], emulsification cross=linking [37,38] and spray drying [39,40,41].

Cetirizine is also an antihistamine, including a neutral antagonist with a similar efficacy to other non-sedative antihistamines such as chlorphenamine, astemizole, terfenadine and loratadine [42]. These second-generation antihistamines are less able to cross the blood–brain barrier and therefore do not cause side effects such as drowsiness and decreased cognitive processing [25]. Cetirizine has an extremely bitter taste. Spray drying is an effective way to mask this taste [43]; for example, Eudragit E PO polymer can be used for this purpose [44].

Cetirizine has a chiral center. The cetirizine on the market is composed of levocetirizine (R-(-)-cetirizine) and dextrocetirizine (S-(+)-cetirizine) [45]. In pharmacology, levocetirizine is an active enantiomer that possesses an affinity for human H1 receptors [46]. Cetirizine has different ionic states at different pHs, of which the neutral/zwitterionic form is the most lipophilic, but nevertheless also hydrophilic [47].

Cetirizine is generally available on the market in the form of tablets, as well as in the form of oral drops. In order to reduce unpleasant symptoms, e.g., an acid or sour stomach, belching, change or loss of taste, indigestion and stomach discomfort [48], as much as possible, it is worth using nasal preparations instead of oral ones. There are nasal fluids, although due to rapid clearance, these flow rapidly through the nasal cavity. The powder-based nasal preparations are more effective, although they are not yet widespread. There are three powder formulations on the market for non-systemic action, including the active ingredients budesonide, beclomethasone dipropionate and dexamethasone cipecilate [49]. In the case of powder-based preparations, the protecting excipients, which can also be encapsulating materials, can successfully help the powder particles stick to the mucous membrane and even promote the absorption of the active ingredient, which should occur as fast as possible. Additives can also act as stabilizers for amorphous powder, preventing crystallization of the active ingredient [50].

Our aim was to prepare dry-powder microcapsules from levocetirizine dihydrochloride (LC) for nasal administration via the nano-spray-drying process. For encapsulation, three different polymers and β-cyclodextrin and its three derivates were chosen. The influence of carriers on powder characteristics and the in vitro release and permeability of the active ingredients on the nasal mucosa was investigated.

## 2. Materials and Methods

### 2.1. Materials

Levocetirizine dihydrochloride was a kind gift from Egis Pharmaceuticals PLC (Budapest, Hungary). Beta-cyclodextrin (β-CD), randomly methylated-beta-cyclodextrin (RAMEB), sulfobutylated-beta-cyclodextrin sodium salt (SBECD) and (hydroxypropyl)-beta-cyclodextrin (HPBCD) were obtained from Cyclolab Ltd. (Budapest, Hungary). Hydroxypropyl methylcellulose (Methocel E5, HPMC) and polyvinyl-pyrrolidone (M_w_ = 40,000) were purchased from Colorcon Ltd. (Budapest, Hungary) and Alfa Aesar (Kandel, Germany), respectively. Polyvinyl alcohol (M_w_ = 30,000–70,000) was a product of Sigma-Aldrich Inc. (Budapest, Hungary). Sodium hydroxide was delivered by Reanal Ltd. (Budapest, Hungary). Isopropyl myristate (IPM) was obtained from Sigma Aldrich (Budapest, Hungary). Phosphate buffer at pH = 7.4 was used as a reference. The PBS was composed of NaCl, KCl, CaCl_2_, Na_2_HPO_4_ and KH_2_PO_4_, and the SNES consisted of NaCl, KCl and anhydrous CaCl_2_ in deionized water, used at pH = 6.0 ± 0.1 with HCl. All these materials were purchased from Sigma-Aldrich (Budapest, Hungary).

### 2.2. Preparation of Composite Microparticles

The drug-loaded composites were manufactured using a Nano Spray Dryer B-90 (Büchi Labortechnik AG, Flawil, Switzerland). Preheated gas was used for drying. The drug and additive solution was pumped through a membrane to produce ultrafine droplets. After the solidification of droplets due to evaporation, an electrostatic collector was used to collect the powder product [51].

The experiments were carried out at a 100 °C inlet temperature and 90 L/min air flow rate with a perforated membrane with 7 µm holes. The spray rate was 30% and the pump rate was 60%. For the experiments, 100 g of solution was prepared as follows: 1 g of LC and 1 g of different carriers (accurate composition of the dried samples can be seen in Table 1) were used, the MilliQ water was weighed, and the solutions were prepared via magnetic stirring and used without filtering. SD9 (HPMC + NaOH) solution (with 1 g of the HPMC carrier) was dried after adjusting the pH to 5.43 with 4 M NaOH (1.46 mL), and NaCl was formed during neutralization of HCl. The names of the dried samples correspond to the type of carrier used (see Table 1), except for SD1 (LC), which did not contain any carrier.

### 2.3. Drug Loading

Drug loading was measured by UV-vis spectrophotometry with a Shimadzu UV-1800 spectrophotometer (Shimadzu, Kyoto, Japan). The absorbance was determined at its maximum value (at a wavelength of 230.5 nm). A total of 0.025 g of LC was dissolved in 100 mL of MilliQ water, and then 5 mL of this solution was diluted to 25 mL to prepare the stock solution. The calibration was achieved using a series of diluted solutions, i.e., 2.5, 5, 10, 20, 30 and 40 µg/mL. The 20 mg product was dissolved in MilliQ water and diluted to 80 µg/mL. The LC content was calculated using Equation (1):(1)$$C_{\mathrm{LC}}\;=\;\frac{(A_{230.5}\;+\;0.011)/0.0321}{80}\;\times \;100$$
where C_LC_ = LC content (%, *w*/*w*) in the dried sample, and A_230.5_ = absorbance at 230.5 nm.

### 2.4. Particle Size Distribution

The particle size analysis was carried out using the laser diffraction method (Malvern Mastersizer 2000, SCIROCCO 2000 Dry Powder Feeder, Malvern Instruments Ltd., Worcestershire, UK). A total of 0.5–1 g of dried product was added to the feeding tray. The dispersion air pressure was 3.0 bar, and the vibration feed was 75%. The following parameters of particle size were included; the volume equivalent diameter values D(4,3), d(0.1), d(0.5) and d(0.9) mark cumulative volume distributions, indicating that 10, 50 or 90% of the particles, respectively, had a size smaller than or equal to the specified size, and span data show the width of the size distribution: d(0.9) − d(0.1))/d(0.5)).

### 2.5. Morphology

An FEI Thermofisher Apreo S (Thermo Fisher Scientific, Waltham, MA, USA) scanning electron microscope was applied for morphology analysis with a 2 kV accelerating voltage.

### 2.6. X-ray Diffraction

X-ray diffraction analysis was carried out using a Philips PW 3710 diffractometer (Philips Analytical, Almelo, The Netherlands) with CuKα radiation, a tube current of 40 mA and a voltage of 50 kV at a scanning rate of 0.02° 2θ/s. Philips X’Pert Data Collector software was used for the measurement control and the data collection.

### 2.7. Viscosity Measurements

Viscosity was measured on an SV-10 Vibro Viscometer (A&D Limited, Tokyo, Japan) at 25 °C.

### 2.8. Thermal Measurements

Thermogravimetry (TG) was achieved using a Setaram LabsysEvo (Lyon, France) TG-DSC system in a high-purity (99.999%) argon atmosphere. Then, 100 μL aluminum crucibles were filled with the samples, and heated from 25 °C to 300 °C at a rate of 10 °C/min. Baseline correction and data processing were carried out with thermoanalyzer software (Calisto Processing, ver. 2.092). The thermal analyzer was calibrated via a multipoint calibration method using seven certified reference materials throughout the whole operating temperature range.

### 2.9. In Vitro Release Study of Nasal Powders

The modified paddle method (USP dissolution apparatus, type II; Pharma Test, Hainburg, Germany) was used to examine the dissolution rate of drug-loaded microparticles. The tests were carried out under nasal temperature and pH conditions. A total of 50 mL of simulated nasal electrolyte solution (SNES) was used as a medium at 32 °C, and the pH was set to 5.6 ± 0.1. In the case of nasal powders, 5 mg of LC-containing microparticles was tested, which was washed in the donor phase, as solid powder. The paddle was rotated at 50 rpm, and the concentration was determined in real time.

Inline measurements of LC content were conducted with an AvaSpec-2048L transmission immersion probe (AVANTES, Apeldoorn, The Netherlands) connected via an optical fiber to an AvaLight DH-S-BAL spectrophotometer (AVANTES, Apeldoorn, The Netherlands). The lack of dilution in the acceptor phase can be a limiting factor in inline measurements, which aim to model the dilution in the blood by transferring the LC from the direct environment of nose.

### 2.10. In Vitro Permeability Test

For the evaluation of LC diffusion through the synthetic membrane, the modified Side-Bi-SideTM (Crown Glass, USA) diffusion test was applied. The cellulose ester membrane (pore diameter = 0.45 µm) was dipped in isopropyl myristate before use. pH = 7.4 ± 0.1 phosphate-buffered saline solution (PBS) was used as an acceptor phase to simulate the blood. A pH 5.6 ± 0.1 was set with HCl solution for the SNES. The temperature was adjusted to 35 °C, and the rate of magnetic stirring was 100 rpm. Preincubation of the membranes in the impregnation agent was carried out for 30 min before the investigations.

A total of 5 mg of each sample was weighed and investigated. The extent of diffusion can be described by the penetration extent (P_LC_, μg/cm^2^) for the nasal formulations.
P_LC_ = (Mass of permeated LC (μg))/(Diffusion surface (cm^2^))

To study the kinetics more thoroughly, we took measurements over 60 min, though the residence time of nasal formulations is ca. 15 min on the nasal mucosa due to the mucociliary clearance [52]. The penetration extent at 15 min was recorded to determine the sequence among the diffused LCs.

The diffusion surface was 0.875 cm^2^, while the area of human nasal mucosa was 160 cm^2^; hence, LC was penetrated more in vivo than in our device. Three parallel measurements were conducted for every sample. Real-time detection was achieved to determine the penetrated LC content at 232 nm using an AvaLight DH-S-BAL spectrophotometer (AVANTES, Apeldoorn, The Netherlands) connected to an AvaSpec-2048L transmission immersion probe (AVANTES, Apeldoorn, The Netherlands).

The flux, J, was calculated using the following Equation (2):(2)$$J=\frac{\partial m}{A\;\partial t}$$
where m is the cumulative mass of LC transported in time t, and A is the membrane surface area. The flux was normalized to the donor concentration (C_d_) in order to calculate the permeability coefficient (K_p_) as follows:
(3)$$K_{p}=\frac{J}{C_{d}}$$


## 3. Results and Discussion

### 3.1. Drug Loading, Yield and Particle Morphology

LC-loaded composite microparticles were prepared using a nano spray dryer for nasal drug delivery. The drying conditions (see Section 2.2) and the concentration of the LC (1% (*w*/*w*)) and carriers (1% (*w*/*w*)) in samples SD1—SD9 were constant. The SD1 (LC) sample was dried without a carrier, while the SD2—SD9 samples were dried with different carriers. Levocetirizine’s ionic state is highly pH-dependent. At pH = 5.43, it exists in a neutral/zwitterionic form [53]. The SD9 (HPMC + NaOH) sample was prepared after neutralization of LC with sodium hydroxide. Amphoteric compounds show the most lipophilic and least soluble characteristics in their neutral form. However, the zwitterions contain the charged parts of a molecule; consequently, they are rarely lipophilic, and they are often soluble in water at any pH [54]. The drug loading of dried microparticles was between 49.1 and 52.9%, i.e., bearing approximately 50% similarity to its ratio to the carrier in the solutions before dying. SD9’s (HPMC + NaOH) drug content was the lowest at 47.2%, because it also contained sodium chloride as a product of neutralization. The lowest yield obtained in the process was 71.1%, for sample SD1 (LC), where the LC was dried without a carrier. The presence of a carrier in the solution increased the process yield to a maximum of 85.1%, which was dependent on the carrier. Detailed data for drug loading and yield can be seen in Table 1.

The particle size of the microparticle composites is very important, taking into account the nasal delivery. Particle sizes of 1–5 µm are suitable for both nasal and pulmonary applications [55]. The bulk LC had an average particle size of 55.94 µm with a bimodal distribution. The SEM image shows a highly agglomerated rod-like crystal morphology (see Figure 1a). The dried LC (SD1 (LC)) had a spherical morphology with an average particle size of 2.42 µm (see Figure 1b). The composite particles had a very similar average particle size and distribution (see Table 1). Their value varied between 2.52 and 3.44 µm. Sample SD7 (PVA) was an exception, having a measured average particle size of 29.08 µm because of its bimodal distribution. According to the SEM images (see Figure 2), the composite particles were spherical, and the individual particle size was similar to that of other composite particles, such as the SD3 (HPBCD), SD4 (RAMEB), SD5 (β-CD) and SD8 (HPMC) samples (see Appendix A). The bimodal distribution may be better explained by inadequate measurement conditions in the dry-powder dispersion unit than the agglomeration of the particles. SBECD is a sodium salt that was neutralized with LC in the solutions. The dried sample contained sodium chloride as well; hence, developed crystals can be observed in Figure 3a. Sample SD9 (HPMC + NaOH) also contained sodium chloride as a result of adding sodium hydroxide to the LC/HPMC solutions. Saline nasal sprays are already available on the market [6] to provide immediate relief of inflamed nasal membranes. In Figure 3b, several small NaCl crystals can be seen on the surface of composite particles. The difference in the appearance of the salt crystals in SD2 (SBECD) and SD9 (HPMC + NaOH) samples can be explained by the different viscosities of the solutions before drying. The diffusion coefficient is inversely dependent on the viscosity. During spray drying, the increase in concentration caused material diffusion in the droplets. The lower the viscosity of a solution, the higher the diffusion coefficient of the substances [56,57]. As a result, in the sample containing SBECD, an additive with low viscosity (0.94 mPa·s), NaCl, diffused easily in the droplet, resulting in larger salt crystals. In the sample with higher viscosity, HPMC (2.13 mPa·s), NaCl mobility was hindered; thus, many smaller crystals were formed.

X-ray mapping showed (Figure 4) that while the raw material was crystalline, the active ingredient and the polymer composites were amorphous in all products. For samples SD2 (SBECD) and SD9 (HPMC + NaOH), some crystalline peaks were also present that were identified as NaCl. The amorphous nature of the products promoted rapid dissolution, which is an important aspect for nasal products.

### 3.2. Evaluation of Dissolution Tests

Based on the hydrophilic character of the drug and applied excipients, 100% of the drug was liberated immediately from the samples. This means that there was no effect of the excipients on the dissolution kinetics. Therefore, despite the encapsulation, the good dissolution behavior of LC was preserved. However, additives could help increase effective drug absorption with a higher concentration of the drug.

### 3.3. Evaluation of Permeability Tests

In terms of their permeability-enhancing effect, cyclodextrins cannot be compared with other penetration enhancers, because they cannot penetrate the skin under normal conditions [58]. Babu and Pandit [59] used side-by-side diffusion cells and phosphate-buffered saline at pH 7.4 in the case of cyclodextrins. The examined HPBCD and partially methylated cyclodextrin were proved to be suitable for improving solubility, and when used in certain concentrations, the penetration of the beta-blocking agent bupranolol was increased.

Our calculated flux and permeability data are summarized in Table 2. The permeability modeling diffusion curve of cyclodextrins, the pure spray-dried LC and LC can be seen in Figure 5. β-CD was found to be the most effective penetration enhancer, probably because it is the least hydrophilic among the applied cyclodextrins. HPBCD and RAMEB minimally improved the penetration of the active ingredient compared to the bulk LC. SBECD achieved an almost similar penetrating effect to that of the bulk drug itself, as also seen with the pure spray-dried active ingredient.

In the case of the other polymer ingredients used (Figure 6), the viscosity played a notable role compared to the LC. The viscosity of 1% (*w*/*w*) of the aqueous solution of PVP, PVA and HPMC was 1.14 mPa·s, 1.44 mPa·s and 2.11 mPa·s, respectively. The diffusion coefficient was inversely proportional to the viscosity of the polymer solutions; hence, PVP improved the penetration the most and HPMC improved it the least. In samples SD6 (PVP), SD7 (PVA) and SD8 (HPMC), the active ingredient was spray-dried in dihydrochloride form; i.e., the pH value of the donor phase may have shifted towards the more acidic range, where the cationic form of the active ingredient might appear, which is more hydrophilic. Comparing the two HPMC-containing powders, i.e., SD8 (HPMC) and SD9 (HPMC + NaOH), the latter one exerted higher penetration than sample SD8 (HPMC). In the SD9 (HPMC + NaOH) sample, the LC was in the zwitterionic form (after neutralizing with hydrochloride to pH = 5.43). Thus, the active ingredient was in the neutral state, in which it is more lipophilic than in the dihydrochloride form [46], which can enhance the penetration. This sample contained NaCl, which could also have played a role in the penetration.

## 4. Conclusions

In the present study, antihistamine levocetirizine dihydrochloride, with different excipients, was nano-spray-dried to produce composite microparticles suitable for nasal delivery. Three mucoadhesive polymers (HPMC, PVA and PVP) and four cyclodextrins (β-CD, RAMEB, SBECD and HPBCD) were used to modify the permeability behavior of LC. The carriers played different roles in the formulations to ensure the stability of the drug, increase the viscosity of the nasal solution after the release of the spray-dried sample and enhance the penetration in the nasal cavity. As a positive control, LC without any carrier was spray-dried. The composite particles had an average particle size between 2.42 and 3.44 µm with the exception of the sample containing PVA, with a particle size of 29.80 µm. The products had spherical morphology according to SEM images and an amorphous state as confirmed by XRD investigations. Two products with NaCl showed some crystalline peaks, and crystals appeared on SEM images. The LC was liberated immediately from the dried products, similar to the bulk LC, which is an important feature, because the nasal mucus undergoes mucociliary clearance every 10–15 min. That is why the enhancement of nasal permeability of a drug is a very important task in addition to promoting the local sustained effects. Based on the permeability test using modified diffusion cells, the flux and permeability coefficients were calculated on the basis of the measured drug diffusion. The flux of bulk LC was 56.96 µg/cm^2^/h, which is very similar to that of the spray-dried pure LC and SD2 (SBECD). SD4 (RAMEB) and SD3 (HPBCD) showed slightly higher permeability. Significantly higher penetration was measured for SD5 (β-CD) (188.37 µg/cm^2^/h). β-cyclodextrin, with the lowest solubility and less hydrophilic character, enhanced the penetration of the LC most substantially. The penetration of HPMC, PVP and PVA composites was lower than that of LC. The measured differences among them could be explained on the basis of their viscosity causing different diffusion efficacies in the nasal fluid.

## Figures and Tables

**Figure 1 pharmaceutics-15-00317-f001:**
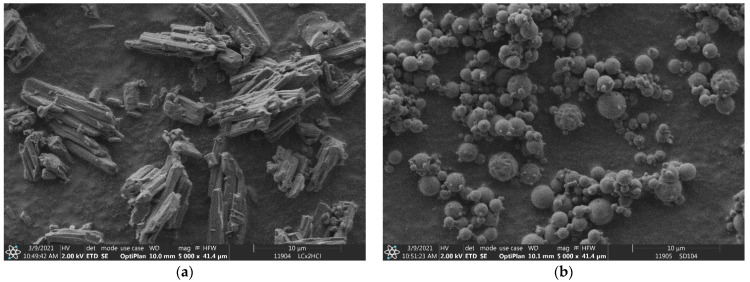
SEM images of bulk LC (**a**) and SD1 (LC) (**b**) samples.

**Figure 2 pharmaceutics-15-00317-f002:**
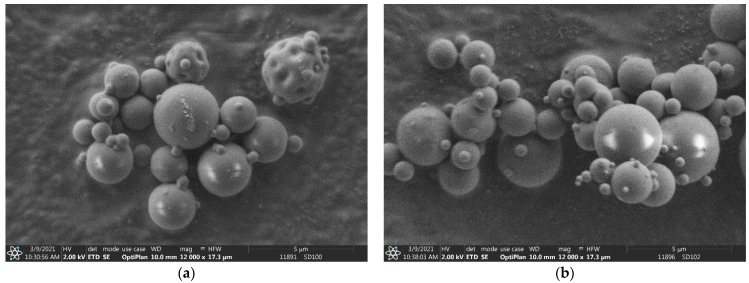
SEM images of SD6 (PVP) (**a**) and SD7 (PVA) (**b**) samples.

**Figure 3 pharmaceutics-15-00317-f003:**
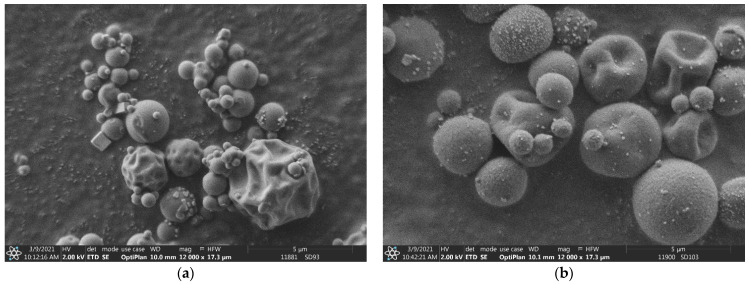
SEM images of SD2 (SBECD) (**a**) and SD9 (HPMC + NaOH) (**b**) samples.

**Figure 4 pharmaceutics-15-00317-f004:**
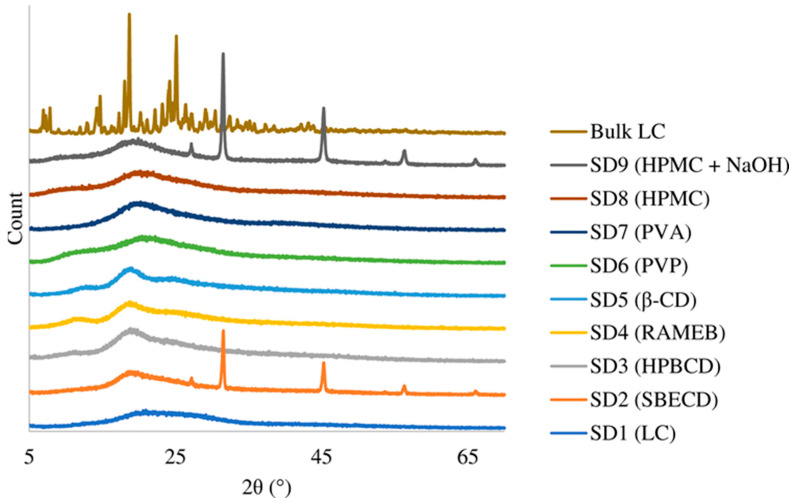
X-ray curves of samples.

**Figure 5 pharmaceutics-15-00317-f005:**
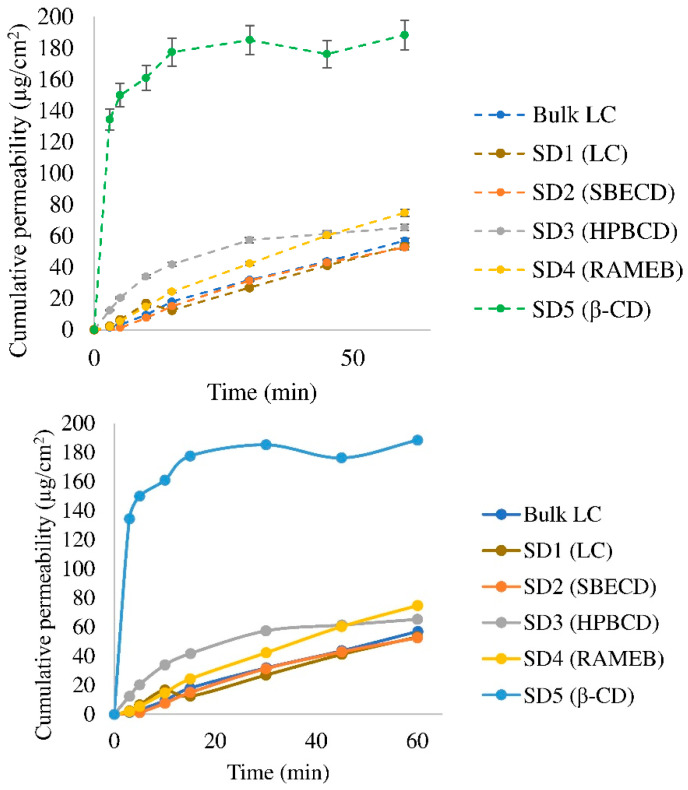
Diffusion curve of bulk LC, the pure spray-dried LC and cyclodextrin composites of LC. (The standard deviation of samples was less than 3%, except for the SD5 (β-CD) sample, for which the standard deviation was less than 5%.)

**Figure 6 pharmaceutics-15-00317-f006:**
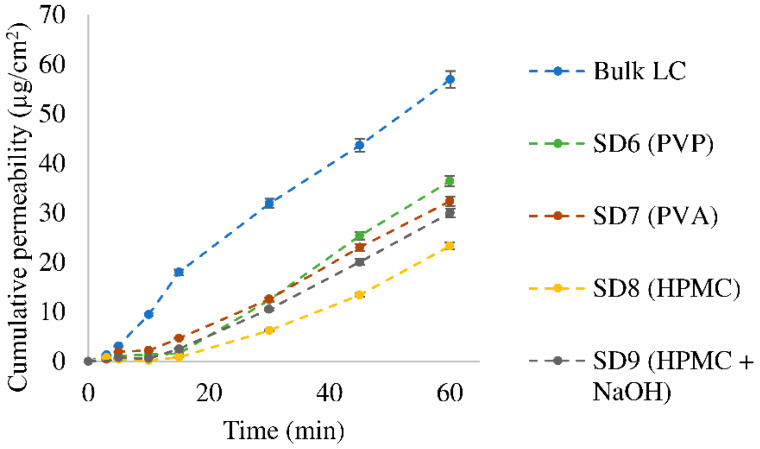
Diffusion curve of bulk LC and its polymer composites. (The standard deviation of the samples was less than 3%).

**Table 1 pharmaceutics-15-00317-t001:** Active ingredient content (C_LC_, %), water content, particle size (D[4,3]) and distribution of the dried products, and yield of the drying process.

No.	Yield (%)	C_LC_, % (*w*/*w*)	Water Content, % (*w*/*w*)	D[4,3] (µm)	d(0.1) (µm)	d(0.5) (µm)	d(0.9) (µm)	Span
SD1 (LC)	71.1	96.2 ^1^	3.8	2.42	1.13	2.16	4.12	1.38
SD2 (SBECD)	78.0	49.1 ± 1.2	3.1	2.82	1.23	2.48	4.92	1.49
SD3 (HPBCD)	71.1	50.5 ± 0.8	2.7	2.52	1.24	2.29	4.14	1.27
SD4 (RAMEB)	79.6	49.9 ± 0.9	2.7	2.88	1.25	2.55	5.01	1.47
SD5 (β-CD)	81.9	52.9 ± 1.2	0.6	2.68	1.21	2.38	4.61	1.43
SD6 (PVP)	83.5	49.9 ± 0.3	4.2	2.74	1.27	2.46	4.62	1.36
SD7 (PVA)	71.8	49.9 ± 0.3	3.1	29.80	1.95	4.72	116.76	24.32
SD8 (HPMC)	82.6	50.0 ± 0.6	2.4	3.09	1.35	2.72	5.37	1.48
SD9 (HPMC + NaOH)	85.1	47.2 ± 0.4	1.6	3.44	1.46	3.03	6.02	1.50
Bulk LC	-	100	-	55.94	2.01	6.16	232.12	37.36

^1^ based on TG.

**Table 2 pharmaceutics-15-00317-t002:** Calculated flux and permeability values.

No.	J (µg/cm^2^/h)	Kp (cm/h)
Bulk LC	56.96	10.3
SD1 (LC)	53.15	9.6
SD2 (SBECD)	52.52	9.5
SD3 (HPBCD)	65.29	11.8
SD4 (RAMEB)	74.73	13.6
SD5 (β-CD)	188.37	35.1
SD6 (PVP)	36.41	6.5
SD7 (PVA)	32.35	5.9
SD8 (HPMC)	23.29	4.2
SD9 (HPMC + NaOH)	29.99	5.4

## Data Availability

Data are included within the article.

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
