# Peer review of "Nano-Spray-Dried Levocetirizine Dihydrochloride with Mucoadhesive Carriers and Cyclodextrins for Nasal Administration"

_pharmaceutics, 2023, doi:10.3390/pharmaceutics15020317_

Round 1

Reviewer 1 Report

The manuscript entitled “Nano spray dried levocetirizine dihydrochloride with mucoadhesive carriers for nasal administration”, authored Mirankó et al., deals with preparation of dry powder microcapsules with levocetirizine dihydrochloride for nasal administration by nano spray drying process. The topic of this manuscript is important and current, and results could be interesting for readers. However, some changes have to be entered into the revised version of the manuscript before it can be further processed:

1.       In line 98 the Authors write "In order to reduce unpleasant symptoms as much as possible..." - please specify what symptoms

2.       Please indicate the qualitative and quantitative composition of each of the prepared formulations

3.       Not all carriers used are mucoadhesive, this should be changed. Or, If the authors wish to leave information on mucoadhesion, studies on this property should be added to the manuscript.

Author Response

Review 1:

  1. In line 98 the Authors write "In order to reduce unpleasant symptoms as much as possible..." - please specify what symptoms

Authors' response: It has been completed in Introduction as follows. „Cetirizine is generally available in the market in form of tablets, but also in the form of oral drops. In order to reduce unpleasant symptoms, e.g. acid or sour stomach, belching, change or loss of taste, indigestion, stomach discomfort [49], as much as possible, it is worth dealing with nasal preparations instead of oral ones.”

  1. Please indicate the qualitative and quantitative composition of each of the prepared formulations

Authors' response: Table 1 contains the compositions; moreover, some information has been added into Preparation of composite microparticles section: „For the experiments 100 g of solutions were prepared as followed: 1 g of LC and 1 g of different carrier (accurate composition of the dried samples can be seen in Table 1.) and the MilliQ water were weighed, the solutions were prepared by magnetic stirring and used without filtering. SD9 (HPMC + NaOH) solution (HPMC carrier was 1 g) was dried after adjusting the pH to 5.43 with 4 M NaOH (1.46 mL), formed NaCl during neutralization of HCl. The name of the dried samples containing the type of used carrier (see Table 1.), except SD1 (LC), that not contained any carrier.”

3.Not all carriers used are mucoadhesive, this should be changed. Or, If the authors wish to leave information on mucoadhesion, studies on this property should be added to the manuscript.

Authors' response: Thank you very much for the comment: The phrase throughout the article was checked and corrected, where it was necessary, including also the title.

Reviewer 2 Report

This manuscript reported the synthesis of dry powder microcapsules from levocetirizine dihydrochloride for nasal administration by nano spray drying process. For encapsulation, three different polymers and four cyclodextrin derivates were chosen. The influence of carriers was investigated on powder characteristics and on in vitro release and permeability of the active ingredients on the nasal mucosa. This manuscript is valuable for publication in Pharmaceutics after the authors address the following concerns.

1.     What are the meanings of D[4,3], d(0.1), d(0.5), d(0.9) and Span in Table 1?

2.     There are ten samples in the manuscript, why only six samples have SEM images?

3.     How many times repeated in Figure 5 and Figure 6? Why are there no error bar in these figures?

4.     Why is the diffusion curve of bulk LC very different from that in Figure 5 and Figure 6?

Author Response

Review 2:

  1. What are the meanings of D[4,3], d(0.1), d(0.5), d(0.9) and Span in Table 1?

Authors' response: It has been added completed in Particle size distribution section. „The following parameters of particle size were included: volume equivalent diameter-D(4,3), d(0.1), d(0.5) and d(0.9) values mark cumulative volume distributions, i.e. 10, 50, or 90% of the particles, respectively, have a size smaller than or equal to the specified size and span data show the width of the size distribution: d(0.9)-d(0.1))/d(0.5)).„

  1. There are ten samples in the manuscript, why only six samples have SEM images?

Authors' response: We believed that the presented SEM images were adequately representative; nevertheless, the SEM images of other four samples have been included in the supplementary.

  1. How many times repeated in Figure 5 and Figure 6? Why are there no error bar in these figures?

Authors' response: In In vitro permeability test section it was mentioned: “3 parallel measurements were done with every sample.”

The error bar was replaced in the images and their captions. „The standard deviation of samples was less than 3%, except for the SD5 (β-CD) sample, where the standard deviation was less than 5%.”

  1. Why is the diffusion curve of bulk LC very different from that in Figure 5 and Figure 6?

Authors' response: The bulk LC curve is the same in the two figures (the y axis size is different), only in Figure 5 a large part is covered by the curve of the SD2 (SBECD) sample, and it could also be confusing that there were two blue curves, so one of them was changed to green.

Round 2

Reviewer 1 Report

I accept manuscript in present form